# An Unconventional Case Study of Neoadjuvant Oncolytic Virotherapy for Recurrent Breast Cancer

**DOI:** 10.3390/vaccines12090958

**Published:** 2024-08-23

**Authors:** Dubravko Forčić, Karmen Mršić, Melita Perić-Balja, Tihana Kurtović, Snježana Ramić, Tajana Silovski, Ivo Pedišić, Ivan Milas, Beata Halassy

**Affiliations:** 1Centre for Research and Knowledge Transfer in Biotechnology, University of Zagreb, 10000 Zagreb, Croatia; dforcic@unizg.hr (D.F.); tkurtovi@unizg.hr (T.K.); 2Centre of Excellence for Virus Immunology and Vaccines, 10000 Zagreb, Croatia; 3Clinical Department of Diagnostic and Interventional Radiology, University Hospital Centre Sestre Milosrdnice, 10000 Zagreb, Croatia; karmen.mrsic@kbcsm.hr; 4Clinical Department of Pathology and Cytology “Ljudevit Jurak”, University Hospital Centre Sestre Milosrdnice, 10000 Zagreb, Croatia; melita.peric.balja@kbcsm.hr (M.P.-B.); snjezana.ramic@kbcsm.hr (S.R.); 5Department of Oncology, University Hospital Centre Zagreb, 10000 Zagreb, Croatia; tajana.silovski@kbc-zagreb.hr; 6Radiochirurgia Zagreb, 10431 Sveta Nedelja, Croatia; ivo.pedisic@radiochirurgia.hr; 7University Hospital for Tumors, University Hospital Centre Sestre Milosrdnice, 10000 Zagreb, Croatia; ivan.milas@kbcsm.hr

**Keywords:** oncolytic virotherapy, measles virus, vesicular stomatitis virus, breast cancer, Edmonston-Zagreb

## Abstract

Intratumoural oncolytic virotherapy may have promise as a means to debulk and downstage inoperable tumours in preparation for successful surgery. Here, we describe the unique case of a 50-year-old self-experimenting female virologist with locally recurrent muscle-invasive breast cancer who was able to proceed to simple, non-invasive tumour resection after receiving multiple intratumoural injections of research-grade virus preparations, which first included an Edmonston-Zagreb measles vaccine strain (MeV) and then a vesicular stomatitis virus Indiana strain (VSV), both prepared in her own laboratory. The intratumoural virus therapy was well tolerated. Frequent imaging studies and regular clinical observations documenting size, consistency and mobility of the injected tumour demonstrate that both the MeV- and VSV-containing parts of the protocol contributed to the overall favourable response. Two months after the start of the virus injections, the shrunken tumour was no longer invading the skin or underlying muscle and was surgically excised. The excised tumour showed strong lymphocytic infiltration, with an increase in CD20-positive B cells, CD8-positive T cells and macrophages. PD-L1 expression was detected in contrast to the baseline PD-L1-negative phenotype. The patient completed one-year trastuzumab adjuvant therapy and remains well and recurrence-free 45 months post-surgery. Although an isolated case, it encourages consideration of oncolytic virotherapy as a neoadjuvant treatment modality.

## 1. Introduction

Breast cancer (BC) is the most diagnosed cancer and the leading cause of cancer-related death in women worldwide [1]. Therapeutic strategies involve surgical removal of the tumour, systemic oncological therapies with curative intent in neoadjuvant and adjuvant settings and a panel of supportive and palliative therapeutic measures in advanced disease. These therapies encompass chemotherapy, radiotherapy, endocrine therapy (for cancers expressing female sex hormone receptors), targeted therapy (predominantly in cases of HER2-positive breast cancer) and more recently immunotherapy, all with varying degree of toxicity, some of which can be even life-compromising. The search for novel therapeutic modalities of equal or greater efficacy and lower toxicity is highly warranted for all breast cancer subtypes, particularly those that are not amenable to targeted or endocrine therapy like triple-negative breast cancers.

Although the first important observations that some viral infections contribute to the reduction in tumours date back almost 100 years, the systematic and focused oncolytic virotherapy (OVT) development in the last 40 years has resulted in the recent acceptance of the first OVT product in clinical practice [2,3]. There are several products targeting BC in clinical development currently [4,5,6]. Despite the plethora of optimistic preclinical data, the translation into clinical usage is obviously facing challenges as the results of clinical studies are often significantly below of what is expected and fail to repeat the preclinical success [3,7]. Clinical studies for each OVT, equally as for any novel cancer treatment, always start in group of patients with metastatic disease who have already passed many rounds of chemotherapy, radiotherapy and/or immunotherapy, and who therefore often have a compromised immune system and overall health status. Recently, OVT development has been directed also towards its use as neoadjuvant therapy before surgery in early-stage cancer patients, and several clinical trials have started, some of which are also targeting breast cancer [8]. 

Here, we present a unique case study describing the usage of OVT for the self-experimental treatment of locally recurrent BC. The therapy utilized attenuated measles and vesicular stomatitis virus preparations that were not of clinical grade, but were made in the patient’s laboratory and used as clarified cell culture supernatants, devoid of extensive purification from host–cell nucleic acids and proteins. Everything described in the manuscript was only feasible due to the unique situation in which a patient was also a virologist with expertise in growing and characterizing human and animal viruses. The case demonstrates the rapid shrinkage and downstaging of the virus-injected tumour mass and may offer important insights to overcome some of the challenges associated with the development of OVT. Namely, sequential administration of different viruses was used to avoid the potential of antiviral immunity, developed during the course of therapy, to suppress viral cytolytic action. Secondly, an intensive schedule of repeated intratumoural (*i.t*.) virus administration was applied to keep the concentration of infective virus in the tumour milieu constantly high and thus enhance its direct antitumour effect. 

## 2. Results

### 2.1. Patient and Case History

The patient was a 50-year-old woman with a history of local recurrence of triple-negative breast cancer (TNBC), as specified in the Appendix A. The tumour was first diagnosed in 2016 with several foci of invasive ductal cancer and was treated by mastectomy followed by adjuvant chemotherapy. In 2018, a small TNBC local recurrence below the suture from the previous mastectomy was surgically removed. However, a small seroma (<1 cm) remained at the site of excision, which was periodically monitored. Phase contrast magnetic resonance imaging (MRI) in 2020 showed that the structure so far described as “seroma” had progressed to a 2 cm diameter solid tumour. It appeared as a hard, palpable, bright red and inflamed nodule with a thin skin above it. Different imaging techniques described it as a circled plate at the base (chest wall side), with a bulging hill at the skin side. MRI, PET-CT scan, and two independent ultrasound estimations all gave matching tumour volume estimations of 2.47 ± 0.06 cm^3^. MRI showed that the tumour had invaded into the pectoral muscle, which was supported by the PET-CT. Skin infiltration was identified on all three of the diagnostic imaging methods. The PET-CT scan showed no evidence of metastatic disease or local spread to regional lymph nodes.

The patient, who is also an expert virologist, anticipating that the recurrent tumour would be of TNBC phenotype for which therapies of only limited efficacies exist, informed her oncologists that she was going to treat this tumour by the *i.t.* administration of viruses similar to oncolytic viruses (that were in clinical development for BC) before undergoing any other treatment. Her oncologists agreed to monitor the progress of the treatment, primarily with the aim of discontinuing the injections and intervening with conventional therapy in the event of adverse effects or tumour progression (which has not occurred). OVT started immediately after all necessary baseline diagnostic tests had been run, including core needle biopsy sampling of the tumour. Histopathological analysis subsequently indicated that the tumour had evolved from TNBC to HER2 3+.

### 2.2. Oncolytic Virotherapy Protocol and Outcomes

OVT consisted of seven MeV applications in three- to four-day intervals over a period of three weeks, followed by three VSV applications separated by two and one week, prior to the surgical excision (Figure 1). Two months after the tumour excision, MeV was applied subcutaneously once around the surgical suture, as a preventive adjuvant treatment. Detailed step-by-step follow up of the OVT is provided in the Appendix A. 

After only two months of therapy, in which a total of 7.89 log CCID_50_ of MeV and 9.07 log CCID_50_ of VSV was administered, the tumour was significantly reduced in size from a baseline volume of 2.47 ± 0.06 cm^3^ estimated by four independent imaging analyses to 0.91 cm^3^, which was the pathologic size of the excised tumour (Figure 1A). The tumour was converted from a hard, fixed nodule with overlying inflamed skin to a much smaller and softer mobile nodule without skin inflammation. At baseline, it was extremely difficult to insert the needle and to inoculate the virus suspension, whereas by the end of therapy, the tumour had softened considerably, enabling easier needle insertion and virus administration. Under ultrasound imaging, the tumour had become less hypoechoic, less spiculated, better circumscribed, and significantly flattened compared to the baseline (Figure 2A). Transient tumour swelling, already described in clinical trials assessing OVT for the treatment of hepatocellular carcinoma [9,10], was noticed at the beginning of the therapy, reaching a maximum volume of 4.28 cm^3^ on day 8. The repeated MRI performed two weeks after the first VSV administration again revealed a transient increase in size (2.17 cm^3^ on day 41), which might have occurred due to the infiltration and multiplication of the VSV-specific lymphocytes. Accordingly, lymph nodes enlarged in both axillae. 

Histopathological analysis of the finally excised tumour confirmed that it was confined to the subcutis, with no infiltration of either skin or pectoral muscle, in contrast to the baseline diagnosis. The tumour showed strong lymphocyte infiltration—45% (in relation to tumour mass) in comparison to baseline 10%, with some areas rich in lymphocytes and fibrous tissue but without discernible tumour cells (Figure 2B). Such a picture of abundant fibrosis is often seen after a complete pathological response to classical neoadjuvant chemotherapy. Immunohistochemical characterization of infiltrating lymphocytes showed that two subpopulations were dominantly increased within the tumour due to OVT: CD20-positive B cells (from 10% to 70%) and CD8-positive T cells (from 30% to 60%) indicating activation of an adaptive immune response (Figure 2C). Infiltration of macrophages (CD68-positive cells) was also increased. In addition, PD-L1 expression was detected after OVT in contrast to the PD-L1-negative phenotype before the treatment with viruses.

The patient had low, but measurable neutralizing antibody titres (NTs) to both viruses before the start of OVT that increased 100 times during the course of OVT and remained high until its ending (Figure 1A).

The patient did not experience any serious side effects during the course of OVT. The needle insertion and suspension administration were tolerably painful at the beginning. The only systemic side effect was experienced after the first application of VSV, manifested as fever and rigors with onset twelve hours after VSV administration, with complete resolution over the next three days.

## 3. Discussion

Oncolytic viruses are first evaluated in patients with advanced cancer stages that cannot be effectively cured by currently existing and approved therapies. Such patients, who have undergone many cycles of chemotherapy and radiotherapy regimes, may be particularly poor candidates for OVT because of their weak immune and overall health status. In these situations, OVT probably works only by direct virus killing of cancer cells and its ability to further impact the cancer through the induction of a cancer-specific immune response might be substantially limited. Early-stage BC has only recently become a target for the development of OVT (registered clinical trials NCT04185311, NCT04102618, NCT02779855), involving T-VEC or pelareorep in combination with chemotherapy/immunotherapy. 

We report an unconventional case of self-experimental OVT in a female patient with locally recurrent BC. Given that the tumour was of epithelial origin, viruses that are known to successfully infect epithelial cells and have documented safety in humans were selected. The Edmonston-Zagreb measles vaccine strain is known for its safety in paediatric vaccines that have been in use for over 40 years [11,12]. Breast carcinomas express abundantly cell surface CD46 and nectin-4, both utilized as cell entry receptors by the vaccine strain of measles, particularly viruses of the Edmonston vaccine lineage [13,14]. Hence, the usage of the Edmonston-Zagreb vaccine strain was found appropriate. In addition, measles OVT has been clinically tested in patients with advanced stages of different types of BC (NCT01846091) [6]. VSV is an animal pathogen, but is almost non-pathogenic for humans, causing flu-like symptoms in the worst case [15]. It was shown to confer protection against re-challenge in a murine model of BC [16]. Wild-type VSV is considered to be potentially neurotoxic to humans [17], based on the strong neurotoxicity observed in mice after intranasal administration, and especially intrathecal and intracerebellar application [18]. Wild-type VSV also induced neurotoxicity in non-human primates, but only after direct intrathecal inoculation [19]. Its neurotoxicity has so far not been demonstrated in any animal species after administration by any other route, especially not *s.c.* Wild-type VSV did not cause any signs of neurological symptoms after intratumoural application in the case presented here. However, despite this, it is essential to be cautious and to further conduct neurotoxicity tests in future safety studies of the VSV-based preparations, especially if developed for the treatment of brain tumours. The strains of MeV and VSV that were used in the current study had not been genetically engineered to improve their oncolytic properties, but were nevertheless effective when administered according to the schedule and protocol described herein. It is important to emphasize that virus preparations were of research grade and of complex biological composition, considering that purification of viruses from host–cell components was not performed. These impurities could also affect the overall outcome of the described protocol. Two cell lines, MRC-5 and Vero, both regulatory acceptable and widely in use for the upstream processing of human viral vaccines [20], were used for the laboratory-grade virus production. MeV was first grown in the MRC-5 cell line, but later we switched to Vero cells because higher titres were achieved, except for the preparation administered on day 19, when MRC-5 was used due to the temporarily unavailable Vero cells. Being aware that high virus titres are needed for successful OVT, the viruses were regularly and repeatedly administered directly into the tumour, with the aim to maintain continuous exposure to infectious viruses in the tumour microenvironment. Further, we anticipated the rapid induction of a virus-specific antibody response, which has also been proven experimentally. Although current knowledge [10] does not imply that anti-viral neutralizing antibodies prevent intratumoural efficacy of OVT, to minimize such a possibility, we exploited a sequential virotherapy strategy in which treatment was initiated using MeV and was switched later to another virus – VSV, after three weeks of therapy. The sequential usage of two different viruses has already been suggested and demonstrated in preclinical models [21,22,23].

CD8-positive T cell and B cell co-localization observed in excised tumours has already been documented and considered as a positive prognostic factor in epithelial ovarian cancers [24,25]. CTLs have been considered the central effectors for optimal elimination of tumour cells [26,27]. The pivotal role of B lymphocytes in anti-tumour immune responses has only begun to be appreciated [28,29]. Since the increase in CD8-positive T cells and CD20-positive B cells was the consequence of applied OVT (Figure 2C), we could speculate that the activated adaptive immune response was directed dominantly toward viral antigens on tumour cells. It is possible that B-cells served as alternative antigen-presenting cells in the tumour environment, sustaining the survival and proliferation of tumour-infiltrating T cells, as already speculated [25,30]. 

The short-term and middle-term outcome of this unconventional treatment, which was devoid of any significant toxicity, was undoubtedly beneficial. The tumour was significantly shrunken in size, and was not infiltrating either muscle below or skin above in contrast to the baseline. Consequently, it was successfully excised. Due to the HER2 3+ phenotype of the excised tumour, the patient additionally completed adjuvant one-year trastuzumab therapy, in line with recommendations for the treatment of HER2 3+ breast cancers [31]. Before this experimental treatment, the patient experienced local recurrences of TNBC twice, in 22- and 21-month intervals, respectively. In contrast, after the OVT, the patient has been in remission for 45 months.

## 4. Materials and Methods

The Edmonston-Zagreb measles vaccine strain (Institute of Immunology Inc., Zagreb, Croatia) [32] and vesicular stomatitis virus Indiana strain (ATCC) were used for laboratory-grade production of viruses for OVT. Viruses were freshly prepared immediately before application and *i.t.* administered, multifocally, in a total volume of 1–2 mL (details in Figure 1B and Appendix A). The course of therapy was monitored by frequent imaging analyses documenting qualitative and quantitative changes in the tumour mass and by quantification of virus-specific antibodies in serum. The final outcomes were assessed by pathohistological and immunohistochemical analysis of the excised tumour in comparison to the core biopsy material taken before the OVT start (details in Appendix A).

## 5. Conclusions

The authors of this isolated and unconventional work clearly state that self-medicating with oncolytic viruses should not be the first approach to dealing with diagnosed cancer, but wish to encourage formal clinical trials of assessing OVT as neoadjuvant therapy in early cancer. The success of the described self-treatment of locally recurrent breast cancer with MeV and VSV preparations supports the possibility that tumour shrinkage and downstaging before surgery may be effectively achieved by OVT, which, if applied in patients with earlier stages of cancers, may also effectively induce anti-tumour immunity. Consequently, better control and long-term outcome of the disease may be achieved.

## Figures and Tables

**Figure 1 vaccines-12-00958-f001:**
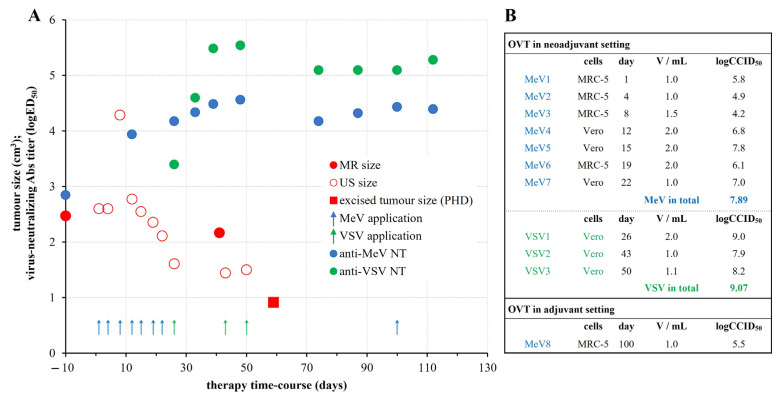
Oncolytic virotherapy (OVT) protocol and outcomes. (**A**) OVT time-course showing timepoints of each individual virus administration (blue arrows for MeV and green ones for VSV) in relation to outcomes monitored as changes in tumour size (red symbols) and virus-specific neutralizing antibody titers (NT) (blue and green circles for anti-MeV and anti-VSV NT, respectively). (**B**) Details on viruses used for each application; CCID_50_-cell culture infective dose 50.

**Figure 2 vaccines-12-00958-f002:**
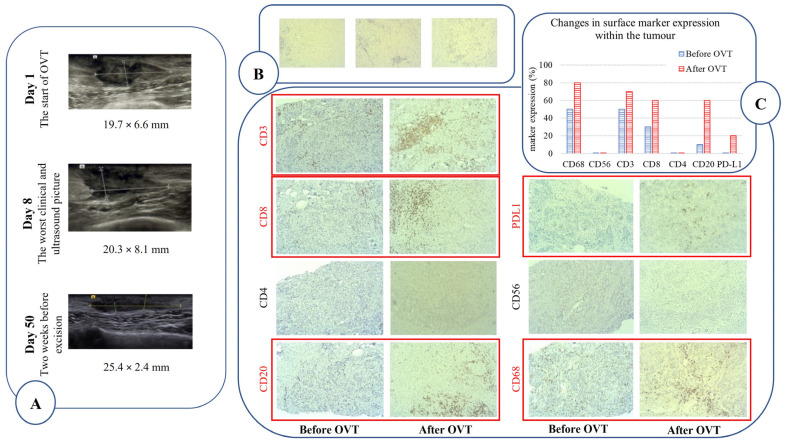
Imaging evidence of the effect of oncolytic virotherapy (OVT). (**A**) Changes in the ultrasound images of the treated tumour made at the same ultrasound system in the same position; representative images with tumour dimensions taken at the beginning and at the end of the therapy, and on day 8 when the worst clinical and ultrasound picture of the tumour size was noted. (**B**) Histopathological analysis of the tumour bed after OVT (three independent tumour sections) shows a histological picture of significant fibrosis, rich in immune cells, and without the presence of malignant cells. (**C**) Immunochemical staining of representative tumour sections before and after OVT, with anti-CD3, anti-CD8, and-CD4, anti-CD20, anti-PD-L1, anti-CD56 and anti-CD68 antibodies. Brown-black dots indicate positive staining. Tumour sections demonstrating an increase in expression of respective surface marker after OVT are denoted in red. Overall changes in each marker expression (in percentages) were evaluated in relation to entire tissue section, except for PD-L1 expression, which was evaluated according to the criteria for triple-negative breast cancer (immune cells only in the intratumoural and peritumoural areas were considered).

## Data Availability

The data supporting the findings reported in this study are available within the manuscript and Appendix A. The tumour samples could be obtained upon reasonable request.

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
