# Peer review of "An Unconventional Case Study of Neoadjuvant Oncolytic Virotherapy for Recurrent Breast Cancer"

_vaccines, 2024, doi:10.3390/vaccines12090958_

Round 1

Reviewer 1 Report

Comments and Suggestions for Authors

see attached file

Comments on the Quality of English Language

English is fine

Reviewer 2 Report

Comments and Suggestions for Authors

The authors provide a case report involving self-treatment (not requiring Ethics committee review) of a 50-YO locally recurrent breast cancer patient with heterologous intratumoral oncolytic virotherapy (OVT) resulting in tumor shrinkage/downstaging and subsequent successful surgical excision and long term DFS (with 1-year maintenance Trastuzumab treatment). Treatment was well-tolerated. Notably the treated tumor displayed profound inflammatory immune cell infiltration and represented a shift from an immunologically “cold” TME at baseline to an immunologically “hot” TME on treatment. These pilot data support potential use of locally delivered OVT as neoadjuvant immunotherapies in the solid (breast) cancer setting. The report is well written, and the results are supported by the data presented.  

My only minor comment for improving the report:

1.)   Given profound B cell infiltration in treated tumors and the focal deposits of T cells and B cells in Fig. 2B imaging results, the authors should explore a brief discussion of tertiary lymphoid structures in a revised Discussion. This is a “hot topic” in the field of immunotherapy currently and a slightly extended discussion would further enrich the report.

Reviewer 3 Report

Comments and Suggestions for Authors

The present case report illustrates an instance of utilizing OVT in the management of recurrent breast cancer. This report demonstrated that OVT exhibits negligible toxicity, and tumor shrinkage and downstaging before surgery might be effectively achieved by OVT. In my opinion, this article is deemed acceptable following revision.

1. The observed side effect thus far is a three-day duration of fever. Can the authors determine if the fever is caused by OVT and whether there are treatment options available during this period?

2. The first paragraph of the full text needs to cite several references to illustrate the advances in breast cancer treatment, the following are recommended references.

[1] J. Li, A. Gu, X.-M. Nong, S. Zhai, Z.-Y. Yue, M.-Y. Li, Y. Liu, Six-Membered Aromatic Nitrogen Heterocyclic Anti-Tumor Agents: Synthesis and Applications. Chem. Rec. 2023, 23, e202300293. DOI: 10.1002/tcr.202300293

[2] Li GL, Li PX, Jiang QY, Zhang QQ, Qiu JR, Li DH, Shan G. Discovery of a pyridophenoselenazinium-based photosensitizer with high photodynamic efficacy against breast cancer cells. Acta Materia Medica. 2023, 2(1): 96-105. DOI: 10.15212/AMM-2023-0002

[3] Shi XY, Bao X, Li Y, Yin CL. Theanine combined with cisplatin inhibits the proliferation and metastasis of TNBC cells through Akt signaling pathway. Tradit Med Res. 2023;8(5):25. doi:10.53388/TMR20221025001

3. The author should meticulously review the syntax of the entire text to ensure its accuracy.

4. The first occurrence of all abbreviations should be accompanied by an interpretation, and they should be consistently used in subsequent descriptions.

5. The authors should meticulously verify the references to ensure their compliance with the journal's stipulations.

Comments on the Quality of English Language

Minor editing of English language required

Round 2

Reviewer 1 Report

Comments and Suggestions for Authors

see comments in rebuttal letter
